# Taphonomy as a Methodological Approach for the Study of Dog Domestication: Application to the Prehistoric Site of Peña Moñuz (Guadalajara, Spain)

**Idoia Claver** [1,2,3], **Verónica Estaca-Gómez** [2,*], **Gonzalo J. Linares-Matás** [4,5], **Jesús Alberto Arenas-Esteban** [6] **and José Yravedra** [2,3,7,*]

1    Department of Geodynamics, Stratigraphy and Palaeontology, Faculty of Geological Sciences, Complutense University of Madrid, Ciudad Universitaria, 28040 Madrid, Spain; idclaver@ucm.es
2    Department of Prehistory, Ancient History and Archaeology, Faculty of Geography and History, Complutense University of Madrid, Ciudad Universitaria, Edificio B, Calle Profesor Aranguren, 28040 Madrid, Spain
3    Research Group 'Quaternary Ecosystems', Department of Prehistory, Ancient History and Archaeology, Faculty of Geography and History, Complutense University of Madrid, Ciudad Universitaria, Edificio B, Calle Profesor Aranguren, 28040 Madrid, Spain
4    St. Hugh's College, University of Oxford, St Margaret's Road, Oxford OX2 6LE, UK; gjl41@cam.ac.uk
5    Emmanuel College, University of Cambridge, St Andrew's Street, Cambridge CB2 3AP, UK
6    Departa History Department, Distance University of Madrid, Carretera de la Coruña, km. 38,500, 28400 Madrid, Spain; jesusalberto.arenas@udima.es
7    C.A.I. Archaeometry and Archaeological Analysis, Department of Prehistory, Ancient History and Archaeology, Faculty of Geography and History, Complutense University of Madrid, Ciudad Universitaria, Edificio B, Calle Profesor Aranguren, 28040 Madrid, Spain
*    Correspondence: vestaca@ucm.es (V.E.-G.); jyravedr@ucm.es (J.Y.)

**Abstract:** The study of early dog domestication has been the focus of considerable scholarly interest in recent years, prompting extensive research aimed at pinpointing the precise temporal and geographic origins of this process. However, a consensus among studies remains elusive, with various research efforts proposing differing timelines and locations for domestication. To address the questions related to the domestication process, researchers have employed a wide range of methodologies, including genetic, biomolecular, morphometric, paleontological, biometric, and isotopic analyses, as well as dental wear analysis to reconstruct paleodiets. Each of these approaches requires access to fossil canid specimens, given that they work directly with the skeletal remains of dogs or wolves. Alternatively, some methods can yield insights into the domestication process without necessitating the physical remains of these canids. Taphonomy, for instance, enables the study of bone surfaces for tooth marks, which may serve as indirect indicators of carnivore activity, potentially attributable to dogs or wolves. This study applies a high-resolution taphonomic analysis to bones modified by carnivores at the prehistoric site of Peña Moñuz. Our aim is to identify the specific carnivores responsible for the observed bone modifications. The findings demonstrate the efficacy of this technique in identifying the agents of bite marks, suggesting that taphonomy may complement the paleogenetic, paleontological, and isotopic methodologies traditionally used to explore the origins of dog domestication

**Keywords:** domestication; *Canis familiaris*; taphonomy; tooth marks; human–animal interactions; geometric morphometrics

## 1. Introduction

Dog domestication has been a focal point of academic inquiry for decades, with numerous studies exploring the origins of this co-evolutionary milestone [1–6]. Most research on

this subject is directed at establishing the precise time and place of domestication. However, the existing studies remain inconclusive, presenting divergent chronologies [4,5,7–18] and distinct geographic origins [3,8,14,19–30], or suggesting parallel domestication events in multiple locations [31].

Identifying the markers of dog domestication is a challenging process. Dogs and wolves, as two morphologically and behaviourally similar canids, present significant challenges for differentiation within the fossil record. To refine these identifications, various methodologies have been developed, including genetics [4,6,11,12,21,32–34], biomolecular approaches [23,35], dental microwear [36], isotopic analyses [4,37,38], biometry [39,40], multi-method approaches [39], and geometric morphometrics [41]. Each of these approaches necessitates the fossil remains of dogs or wolves, an often-challenging requirement given the limitations of the fossil record and the variability in site preservation conditions. Conversely, other methods that rely on indirect evidence can provide insights into canid activity, thereby contributing valuable information on the domestication process.

In recent years, utilizing a taphonomic approach, certain studies have classified tooth marks made by different carnivores with a high degree of probability [42–46]. Research involving modern samples created by dogs and wolves has achieved considerable success in identifying which specific canid left the marks [42–46]. Drawing on this approach, the present study applies these methods to an archeological sample from the prehistoric site of Peña Moñuz (Guadalajara) to evaluate whether the current techniques for distinguishing wolf and dog tooth marks are suitable for application to archeological samples.

The rationale for considering this technique as a potentially valuable tool for studying tooth marks is as follows:

1.  Many prehistoric sites with well-preserved faunal remains feature numerous bones bearing tooth marks. If dogs were present at these sites, it is plausible that they created some of these marks. Identifying the origin of such marks could provide indirect evidence of the actions of dogs in these contexts.

2.  Testing the hypothesis outlined above would be challenging without a reliable method to classify tooth marks. However, previous research has demonstrated that tooth marks made by different predators can be classified with high accuracy [42–46]. Moreover, studies suggest that tooth pits created by different individuals of the same carnivorous species often exhibit consistent patterns. For example, observations of wolves from distinct populations—such as Flechas and Villardeciervos in Zamora, Cabárceno in Cantabria, and Hosquillo in Cuenca—show that both captive and wild individuals produce similar tooth pits [47]. Similarly, research on other carnivores, including leopards and tigers, indicates that individuals of the same species, even with significant size differences due to sexual dimorphism, leave comparable tooth marks. For instance, male and female leopards, as well as tigers, produce similar patterns of tooth pits [48].

To evaluate the resolution and utility of this analytical approach, we conducted a preliminary test in this study. We analyzed tooth marks from a prehistoric site to determine which carnivores were responsible for the marks and to assess the method's applicability to the fossil record.

## 2. The Site of Peña Moñuz

Peña Moñuz is a settlement spanning 4600 m$^2$, located in the town of Olmeda de Cobeta in Guadalajara (Figure 1). It sits on a high and prominent limestone platform within the Dehesa de Olmeda, with a nearly vertical escarpment at an elevation of 1240 m above sea level. The site occupies a strategic point in the landscape and is characterized by a complex defensive architectural layout [49,50]. Peña Moñuz lies within the geomorphological

context of the Upper Tagus basin, where fluvial dissection has eroded the terrain to form sharp valleys bordered by limestone ridges, ultimately creating karst landscapes [51]. Chronologically, the site was occupied over several phases, with notable habitation during the Iron Age (Table 1), particularly between the 4th and 2nd centuries BCE [52].

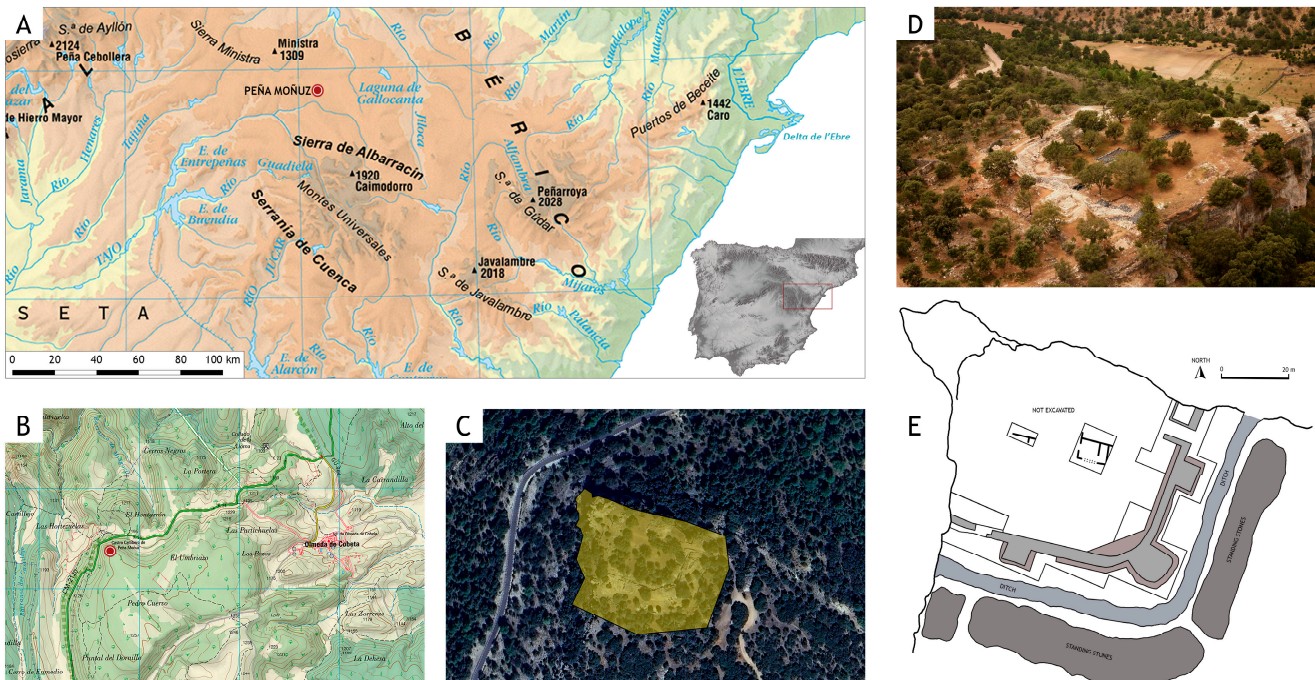

**Figure 1.** (**A**) Physical map of Spain, 1:3,000,000 scale, by the National Geographic Institute (IGN). (**B**) National Topographic Map (1:50,000 scale) Sheets 0488 (Ablanque) and 0489 (Molina de Aragón), by the IGN. (**C**) Satellite image from Google Earth with a cadastral view (parcel 378, polygon 004). (**D**) Panoramic photograph of the Peña Moñuz site from the east. (**E**) Plan of the prehistoric settlement at Peña Moñuz, modified from Arenas-Esteban, J. (see [50]).

**Table 1.** Radiocarbon dating ($^{14}$C) samples from Peña Moñuz (see [52]).

| Lab Reference | Material | Radiocarbon Date | Average Date | Calibrated Dates |
|---|---|---|---|---|
| Beta 270931 | Bone | 2290 ± 40 BP | 2210 ± 40 BP | 400-350 BC/290-220BC |
| Beta 270932 | Bone | 2180 ± 40 BP | 2100 ± 40 BP | 370-150 BC/140-110BC |
| Beta 270933 | Bone | 2220 ± 40 BP | 2140 ± 40 BP | 390-180BC |
| Beta 270934 | Bone | 2290 ± 40 BP | 2190 ± 40 BP | 400-350 BC/290-220BC |
| Beta 99068 | Bone | 2280 ± 40 BP | 2210 ± 40 BP | 400-355 BC/265-230BC |

Since 2006, archeological work led by J. Arenas Esteban has defined multiple functional areas within the site dedicated to activities such as milling, metalworking, and food storage. Zooarcheological studies of Peña Moñuz indicate a faunal accumulation dominated by domestic animals, primarily caprines, cattle, and pigs. Other domestic species, such as horses and dogs, are also represented at the site, although their remains are relatively scarce [53]. Mortality patterns reveal a predominance of adult individuals across all the species, suggesting that caprines and cattle were likely exploited for milk or wool production. Once these animals reached the end of their productive lives, they were slaughtered and used for meat, as evidenced by bones with cut marks. Additionally, the presence of various tooth marks on the bones suggests they were scavenged by carnivores, likely dogs—a hypothesis this study seeks to test by applying high-resolution taphonomic analyses.

## 3. Materials and Methods

The primary objective of this study is to analyze the tooth marks found at Peña Moñuz to identify the carnivores responsible for creating them. To achieve this, we applied high-resolution taphonomic techniques and compared our findings with a comprehensive database of tooth marks produced by various carnivores.

Carnivores leave a range of distinct tooth marks, including "pits" (indentations created by tooth pressure on bone), "scores" (scratches from teeth dragging across bone), "punctures" (perforations on bone surfaces), and "holes" (larger openings created by digestive processes) [54–66].

Our sample consisted of 26 pit marks found on 15 long bone shafts of caprines, bovines, and suids from Peña Moñuz (Figure 2). The distribution of the tooth marks analyzed in this study included 8 bones with a single pit, 5 bones with 2 pits, 1 bone with 3 pits, and 1 bone with 5 pits. We focused specifically on pits because these are the only marks that exhibit consistent characteristics whether created by captive or wild carnivores [43–45,48]. In contrast, other marks, like scores, tend to vary slightly depending on whether the carnivores are in captivity or the wild [44].

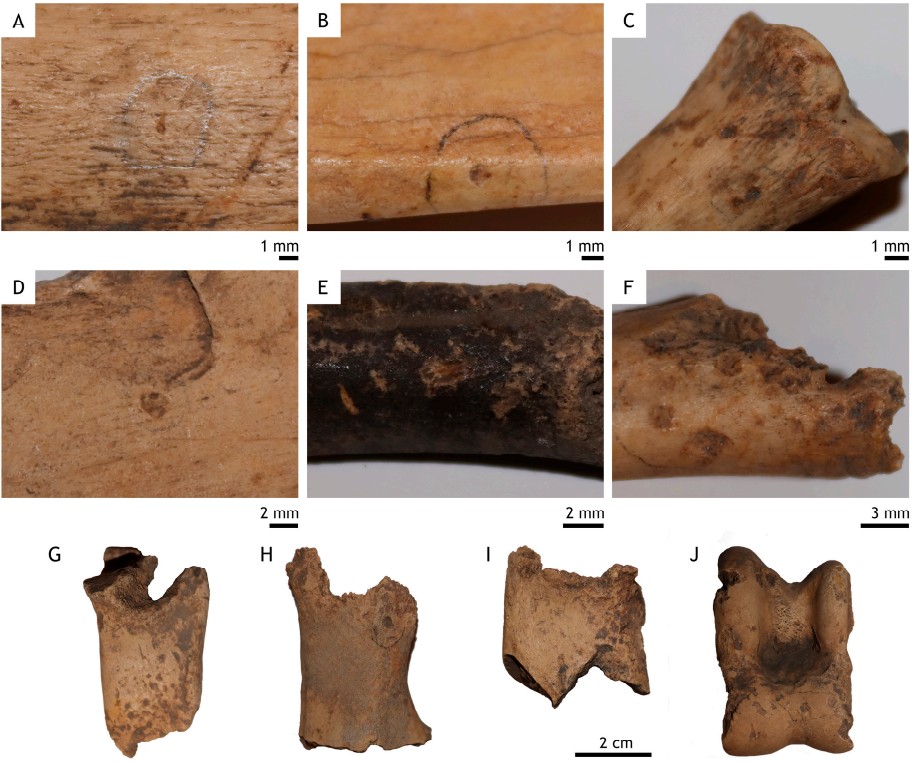

**Figure 2.** Examples of bones with tooth marks at Peña Moñuz. (**A–D**) Pits ranging from 0.1 to 0.2 cm on long bones. (**E**) Multiple pits on burned bones. (**F–I**) Examples of furrowing on long bones. (**J**) Pits on astragalus bone.

The comparative database included pits generated by various carnivores from the subfamilies Caninae (*Canis lupus signatus*, *Canis familiaris*, *Lycaon pictus*, and *Vulpes vulpes*), Ursinae (*Ursus arctos*), Hyaeninae (*Crocuta crocuta*), and Pantherinae (*Panthera leo*, *P. onca*, and *P. pardus*) (see [42–44,67,68]). We also incorporated pits from six different dog breeds, as described in [46]. In total, 658 tooth marks were analyzed, averaging 50 pits per carnivore, with the following distribution for the dogs: Mastiff (50 pits), Boxer (28 pits), mixed breed (50 pits), Labrador retriever (50 pits), Rottweiler (50 pits), and Irish Setter (30 pits).

Following the methodologies presented in previous studies [44,47,67–69], only pit tooth marks were used for comparative analysis, as score tooth marks are subject to greater

variability (see discussion in [47]). The comparative samples used for Peña Moñuz have been well documented in earlier works [42–46]. In all the cases, the tooth marks analyzed were located on the diaphyses of long bones and were produced by at least two individuals of each carnivore species [44]. The samples were obtained through collaborations with various natural parks.

The Parque de la Naturaleza de Cabárceno provided access to horse diaphyses processed by wolves, African wild dogs, lions, jaguars, brown bears, and spotted hyenas [44]. Biopark Fuengirola (Málaga) supplied samples modified by leopards [44,48]. The Hosquillo Park contributed wolf-modified samples from prey such as goats and deer [44,45,67,68]. Finally, tooth marks produced by wild foxes were collected from sheep bones in Ayllón (Segovia) [69].

Most of the analyzed samples come from horse long bone diaphyses, but bones from smaller animals, such as goats, wild boar, and deer, were also included [69]. Despite these differences, previous studies have shown that prey size does not significantly affect the morphology of tooth marks [44,67], enabling comparisons across different prey types. The samples analyzed for each carnivore are summarized below and detailed in [44]:

Brown Bears (*Ursus arctos*): A total of 50 pits from horse diaphyses collected in the summer of 2020 at Cabárceno Park. The marks were produced by multiple adult individuals of varying sexes.

Spotted Hyenas (*Crocuta crocuta*): A total of 50 pits from horse tibiae and radii collected in 2012 at Cabárceno Park, produced by a single adult female. The bones exposed for extended periods were typically consumed entirely, so these samples were retrieved after only a few hours.

Wolves (*Canis lupus*): A total of 50 pits from both wild and captive individuals, collected from Villardeciervos and Flechas (wild populations, Zamora, Spain) and the Hosquillo Park (Cuenca, Spain) and Cabárceno Park (Cantabria, Spain). A total of 571 tooth marks were recorded on horse, ibex, deer, and boar limb bones, from which 50 were randomly selected for analysis. Bone collection occurred over several days to weeks after feeding.

Foxes (*Vulpes vulpes*): A total of 50 pits from sheep long bone diaphyses collected in 2002 at Ayllón (Segovia). Details are available in [69].

African Wild Dogs (*Lycaon pictus*): A total of 50 pits from horse radii and tibiae collected in summer 2010 at Cabárceno Park, produced by two individuals of both sexes. Further details can be found in [70].

Jaguars (*Panthera onca*): A total of 50 pits from horse radii and tibiae collected between 2009 and 2010 at Cabárceno Park, created by multiple individuals of varying sexes [71].

Leopards (*Panthera pardus*): A total of 50 pits from Sri Lankan leopards (*P. p. kotiya*) at Biopark Fuengirola (Málaga). These samples, collected in autumn 2020, include cow axial skeletal bones [48].

Lions (*Panthera leo*): A total of 50 pits from horse radii and tibiae collected in 2011 at Cabárceno Park, produced by several individuals of varying sexes.

Domestic Dogs (*Canis familiaris*): The samples included medium to large breeds (e.g., Spanish Mastiff, Boxer, mixed-breed, Labrador retriever, Rottweiler, and Irish Setter), with 28–50 pits analyzed per breed. These breeds were selected for their similarity to wolves in tooth mark characteristics [46].

The experimental protocols entailed the following steps:

1. Carnivores were provided with whole horse bones (typically radius and tibiae).
2. The bones were semi-defleshed before being offered to the animals after feeding.
3. The bones were left in enclosures for several days.

4. The bones were subsequently collected and boiled in water to preserve tooth marks, avoiding chemical cleaning agents that could degrade them. After boiling and drying, the bones were ready for examination.

5. Pits on each bone were identified and located in preparation for further comparative study in the subsequent stages of the research.

To analyze the tooth marks, we followed the established methodologies (see [42–46,67,68]), which emphasize comprehensive three-dimensional documentation of pits using geometric morphometrics and robust statistical analysis. These results were then compared with the reference dataset to account for multiple carnivore species. This approach overcomes the limitations of earlier metric-based analyses of tooth marks [72].

The first step in this process was to identify pit marks, which were then scanned using a DAVID SLS-2 structured light 3D scanner (Figure 3) from the Archaeometry and Archaeological Analysis Unit at the Research Support Center for Earth Sciences and Archaeometry at the Complutense University of Madrid. Macro lenses ranging from 2x to 10x magnification were fitted to both the camera and the projector. Before scanning, the equipment was calibrated using a marker template with 15 mm intervals between points. The results were saved in .obj format and converted to .ply format for processing in the landmark delimitation software.

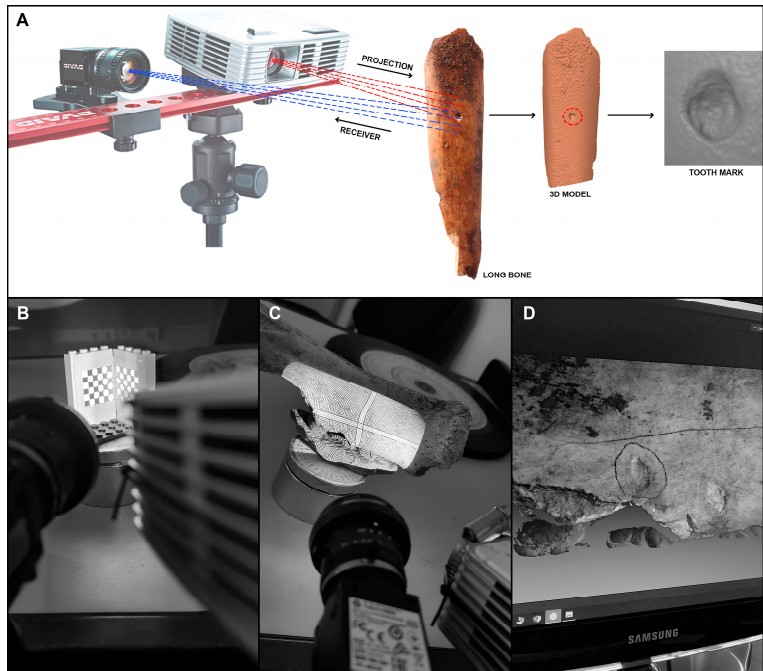

**Figure 3.** (**A**) Digital sample generation using the 3D DAVID SLS-2 scanner. (**B**) A 15 mm marker template. (**C**) Projection by the projector and camera. (**D**) Virtual model of the tooth mark.

Once scanned, and following the protocol by Courtenay et al. (see [43]), 25 landmarks were defined for each pit. These included four landmarks outlining the mark's perimeter, one marking the deepest point, and 20 others distributed across the entire morphology of the mark. The landmarks were consistently placed using the Landmark software (version 3.0.0.6), and the data were exported in Morphologika text format for a morphometric analysis [73,74].

To analyze and compare the data, we performed statistical analyses including Principal Components Analysis (PCA) and Multivariate Analysis of Variance (MANOVA) using the R programming language and relevant functions from the "shapes" and "RVAideMemoire" packages.

First, a Generalized Procrustes Analysis (GPA) was conducted to optimally align the landmarks through translation, rotation, and scaling [75–79]. This type of analysis extracts morphological data from the initial samples, producing configurations based on "shape" (with scaling) and "form" (without scaling). In this case, however, scaling was unnecessary as the sample size did influence comparative outcomes. Thus, we focused solely on original-scale shapes for the GPA using the function procGPA (x$coords, scale = FALSE). This analysis yields Principal Component (PC) scores, which can then be plotted via the "data.frame" and "ggplot" functions to produce a PCA graph.

Lastly, Multivariate Analysis of Variance (MANOVA) was used to compare two samples to determine similarity. This was implemented using the "pairwise.perm.manova" code. To select an appropriate test (either "Hotelling-Lawley" or "Wilk's Lambda"), we assessed whether the PC scores followed a normal (Gaussian) distribution, verified through the Shapiro–Wilk test. The MANOVA results showing $p$-values < 0.003 [44,68] indicate a significant similarity between the compared samples.

Additionally, we included allometric analyses to explore shape–size relationships. Using the "geomorph" package [73], we applied the following functions: "gpagen" for Generalized Procrustes Analysis (GPA), "geomorph.data.frame" to create a dataframe that preserves the sample size (unscaled), "procD.lm" to conduct Procrustes ANOVA (taking centroid size and sample type into account) with a unique allometric formula **(shape ~ log(Csize) * group, data = dataset), and "plotallometry"** to observe shape–size covariation using the "PredLine", "RegScore", "size.shape", and "CAC" methods.

## 4. Results and Discussion

In analyzing the pits from Peña Moñuz, we conducted an initial comparative analysis with tooth marks produced by other carnivores, including felines, ursids, and other canids. This comparison allowed us to evaluate the relationship between the pits at Peña Moñuz and the reference framework established for various carnivores.

Our initial Principal Component Analysis (PCA), presented in Figure 4, demonstrates that the tooth pits from Peña Moñuz are relatively smaller compared to those produced by other carnivores, including foxes and wolves (Figure 4). To provide more robust results, we applied Multivariate Analysis of Variance (MANOVA) using the Wilks' Lambda method. The preliminary Shapiro–Wilk test demonstrated a non-homogeneous distribution, justifying this choice. The MANOVA documented a similarity between the Peña Moñuz values and those for *Canis familiaris*, with a $p$-value of 0.023 (Table 2). Conversely, comparisons with other carnivores yielded significantly lower $p$-values (all $p = 0.001$). Given that the comparison with dogs exceeds the 0.003 threshold ($3\sigma$), this suggests that the tooth marks at Peña Moñuz likely correspond to dogs.

Based on these findings, the tooth marks identified at Peña Moñuz show alignment with those produced by dogs. To strengthen these conclusions, additional PCA and MANOVA analyses were conducted specifically comparing the Peña Moñuz samples to those of *Canis familiaris* (Figure 4). In the PCA, the Peña Moñuz ellipse is contained within the *Canis familiaris* ellipse. For MANOVA, the Wilks' Lambda method yielded a $p$-value of 0.023, confirming that the samples are statistically similar. Thus, there appears to be a relationship between the tooth pits from Peña Moñuz and the marks typically associated with dogs (Table 2).

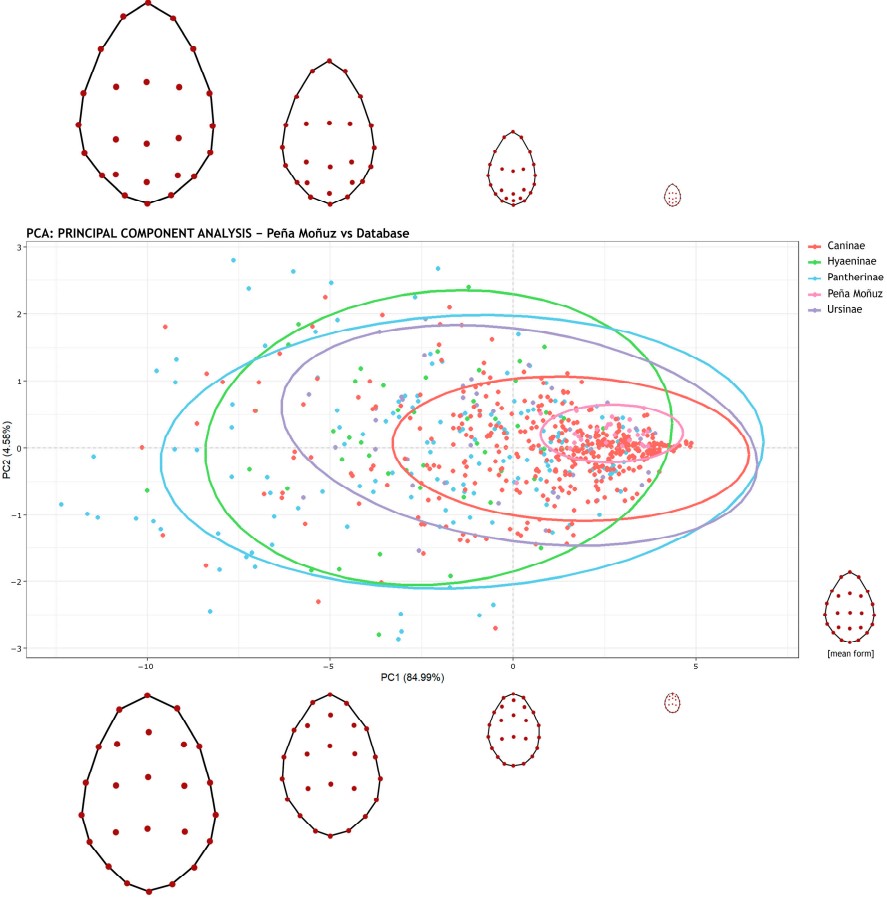

**Figure 4.** Principal Components Analysis (PCA) showing the PC1 and PC2 values of the tooth marks of Peña Moñuz alongside data from other animals in the database grouped by subfamilies, with a distribution of landmarks and size (form). The data presented in this analysis were not scaled (scale = FALSE), ensuring that size and form were taken into account.

**Table 2.** Results of the *p*-values from the Multivariate Analysis of Variance (MANOVA) comparing the Peña Moñuz values to those of other carnivores and humans, as determined using the Wilks' Lambda method. The calculations considered the size of the samples and, consequently, the size of the individuals. In other words, the tooth marks were not scaled (scale = FALSE).

| | *U. arctos* | *C. familiaris* | *V. vulpes* | *H. sapiens* | *C. crocuta* | *P. onca* | *P. pardus* | *P. leo* | *L. pictus* | Peña Moñuz |
|---|---|---|---|---|---|---|---|---|---|---|
| *C. familiaris* | 0.001 | - | - | - | - | - | - | - | - | - |
| *V. vulpes* | 0.001 | 0.002 | - | - | - | - | - | - | - | - |
| *H. sapiens* | 0.068 | 0.174 | 0.052 | - | | | | | | |
| *C. crocuta* | 0.002 | 0.001 | 0.001 | 0.005 | - | - | - | - | - | - |
| *P. onca* | 0.001 | 0.001 | 0.001 | 0.023 | 0.642 | - | - | - | - | - |
| *P. pardus* | 0.002 | 0.002 | 0.017 | 0.046 | 0.001 | 0.001 | - | - | - | - |
| *P. leo* | 0.001 | 0.001 | 0.001 | 0.001 | 0.453 | 0.755 | 0.001 | - | - | - |
| *L. pictus* | 0.052 | 0.001 | 0.001 | 0.071 | 0.434 | 0.260 | 0.004 | 0.022 | - | - |
| Peña Moñuz | 0.001 | **0.023** | 0.001 | 0.001 | 0.001 | 0.001 | 0.001 | 0.001 | 0.001 | - |
| *C. lupus* | 0.072 | 0.003 | 0.001 | 0.667 | 0.001 | 0.001 | 0.003 | 0.001 | 0.004 | 0.001 |

Despite these results, we conducted a second test comparing the tooth marks from Peña Moñuz with those produced by other canids, including foxes, wolves, and various dog breeds (Figure 5). Additionally, to rule out the possibility of human activity, we also compared the Peña Moñuz marks with those associated with human tooth marks (Table 2, Figure 5).

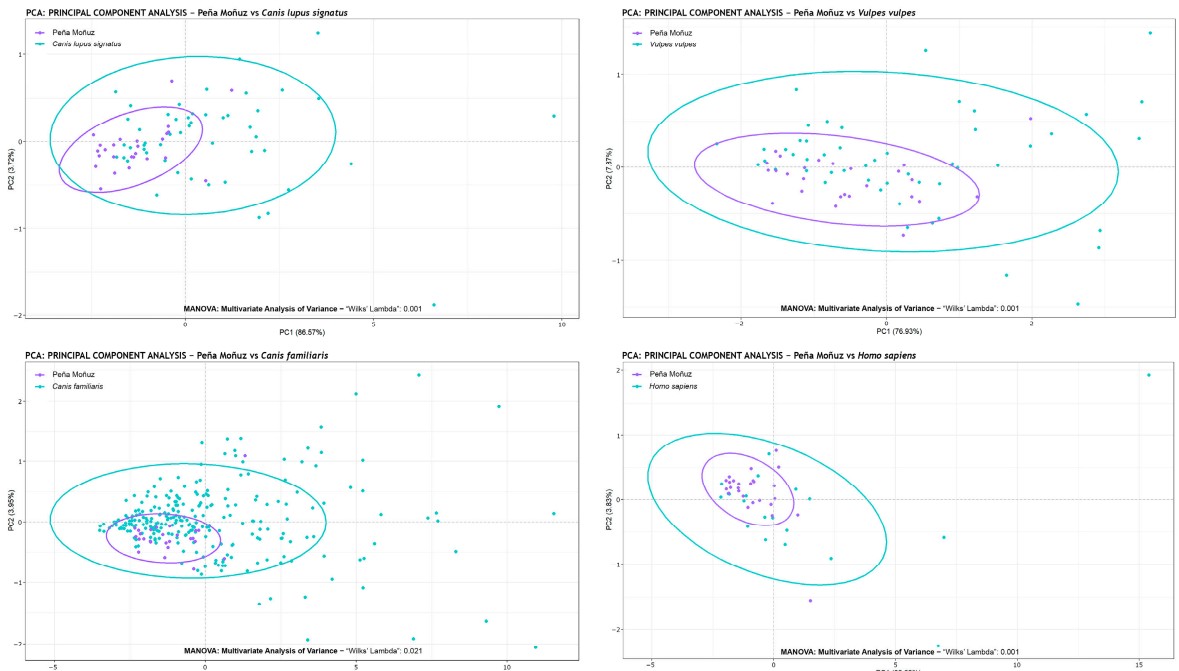

**Figure 5.** Principal Components Analysis (PCA) showing the PC1 and PC2 values of the tooth marks of Peña Moñuz and some canids, *Canis lupus signatus*, *Canis familiaris*, and *Vulpes vulpes*, and the experimental of *Homo sapiens*. The data presented in this analysis were not scaled (scale = FALSE), ensuring that size and form were taken into account. Also, the MANOVA results using the Wilks' Lambda method.

Our findings indicate that the tooth marks from Peña Moñuz show statistical similarity exclusively with dogs. Comparisons with wolves, foxes, and humans yielded *p*-values below 0.003, with all three cases showing a *p*-value of 0.001. In contrast, the values associated with *Canis familiaris* were *p* = 0.021 (Figure 5).

Having established that the Peña Moñuz tooth marks are related to dogs, we then sought to identify the specific type of dog that may have produced these marks (Figure 6, Tables 3 and 4). The MANOVA results in Tables 3 and 4 indicate that whether scaling is applied (scale = TRUE) or not (scale = FALSE), the Peña Moñuz tooth pits align most closely with those produced by Labrador retriever (Figure 6). However, this does not imply that a Labrador made these marks, as this breed did not exist during the Iron Age. Instead, it suggests that the tooth pits from Peña Moñuz could have been created by a canid with characteristics similar to those of a Labrador. Specifically, this points to a dog weighing approximately 25 to 35 kg.

**Table 3.** Multivariate Analysis of Variance (MANOVA) showing the *p*-values of the tooth marks of Peña Moñuz and the different dog breeds using the Wilks' Lambda method. The data presented in this analysis were scaled (scale = TRUE), and size was not taken into account.

|  | Boxer | Labrador | Mastiff | Mixed Breed | Peña Moñuz | Rottweiler |
|---|---|---|---|---|---|---|
| Labrador retriever | 0.001 | - | - | - | - | - |
| Spanish Mastiff | 0.001 | 0.001 | - | - | - | - |
| Mixed breed | 0.001 | 0.001 | 0.223 | - | - | - |
| Peña Moñuz | 0.001 | **0.007** | 0.001 | 0.001 | - | - |
| Rottweiler | 0.050 | 0.001 | 0.001 | 0.001 | 0.001 | - |
| Irish Setter | 0.026 | 0.001 | 0.001 | 0.001 | 0.001 | 0.001 |

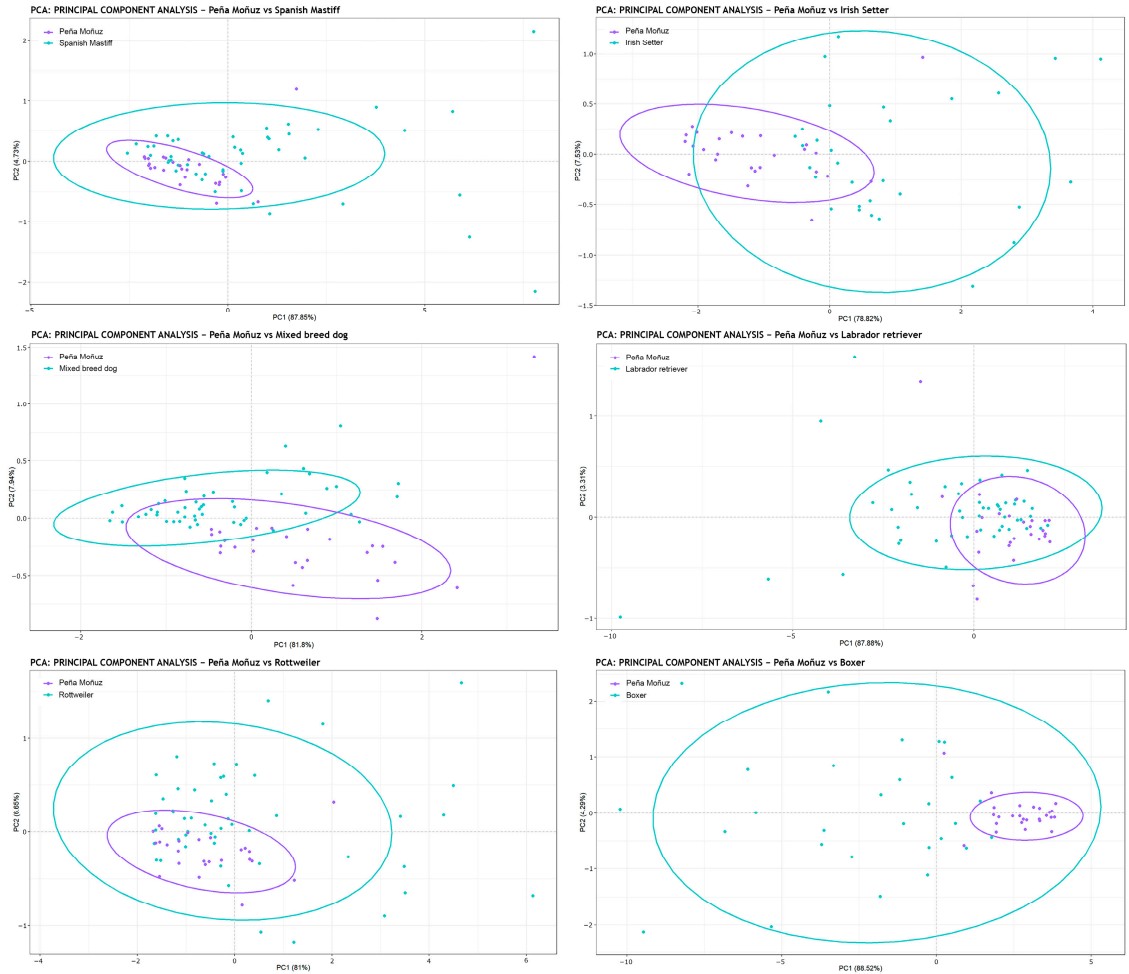

**Figure 6.** Principal Components Analysis (PCA) showing the PC1 and PC2 values of the tooth marks of Peña Moñuz with the different dog breeds and wolves. The data presented in this analysis were not scaled (scale = FALSE), ensuring that size and form were taken into account.

**Table 4.** Multivariate Analysis of Variance (MANOVA) showing the *p*-values of the tooth marks of Peña Moñuz and the different dog breeds using the Wilks' Lambda method. The data presented in this analysis were not scaled (scale = FALSE), ensuring that size and form were taken into account.

|  | Boxer | Labrador | Mastiff | Mixed Breed | Peña Moñuz | Rottweiler |
|---|---|---|---|---|---|---|
| Labrador retriever | 0.001 | - | - | - | - | - |
| Spanish Mastiff | 0.001 | 0.001 | - | - | - | - |
| Mixed breed | 0.001 | 0.001 | 0.001 | - | - | - |
| Peña Moñuz | 0.001 | **0.004** | 0.001 | 0.001 | - | - |
| Rottweiler | 0.001 | 0.001 | 0.001 | 0.001 | 0.001 | - |
| Irish Setter | 0.001 | 0.001 | 0.001 | 0.001 | 0.001 | 0.001 |

This finding is significant because prior research has proposed the possibility of Celtiberians using dogs resembling the modern Spanish Mastiff [80]. However, our study, along with osteometric measurements from dogs at other archeological sites dating from the Late Bronze Age, Iron Age, and early Roman period on the Iberian Peninsula, suggests that medium-sized dogs were more common. When shoulder height could be estimated, most canids measured between 30 and 50 cm at the withers, which corresponds to the dimensions of medium-sized dogs (Supplementary Table S1).

## 5. Conclusions

The findings from this study indicate that the methods used to classify tooth marks successfully associated the pits at Peña Moñuz with dog tooth marks. These results align with expectations, as dog remains have been found at Peña Moñuz [53]. However, it is noteworthy that tooth marks from other carnivores—such as fox, wolf, or badger—found at this and other nearby sites are absent from this sample [53]. Consequently, this study demonstrates that the methodology applied here may serve as an effective approach for documenting canid presence in earlier prehistoric contexts, complementing other established methods, including genetics [4,6,11,12,19,23,27,41], morphometric and biometric paleontological analyses [3,7,39,40], paleodietary analyses [4,36–38], or others.

The present study opens new avenues for future research. Expanding the range of dog types studied would strengthen this line of analysis. While this study's primary aim was to demonstrate the effectiveness of the morphometric analysis of tooth marks in exploring dog domestication, future research could pursue the identification of the specific dog types responsible for bone markings. Currently, this is beyond our reach due to insufficient data on the characteristics of dogs at Peña Moñuz and its surroundings. However, through the analysis of tooth marks, we have been able to rule out the possibility that these marks were produced by large dogs, such as Mastiffs. Instead, we have determined that the tooth marks at Peña Moñuz were made by medium-sized dogs.

In conclusion, while the results from this study are promising, further steps are needed as follows: (1). expanding the current dog databases to enhance the robustness of studies on tooth marks produced by dogs, and (2). analyzing additional prehistoric sites to test the feasibility of identifying dog activity in older contexts.

**Supplementary Materials:** The following supporting information can be downloaded at https://www.mdpi.com/article/10.3390/heritage8010034/s1, Supplementary Materials Table S1 [81–84].

**Author Contributions:** All the authors have contributed to the production of this text at different times during the research. Conceptualization: J.Y., I.C. and V.E.-G.; methodology and software: I.C.; validation: G.J.L.-M.; formal analysis: V.E.-G. and G.J.L.-M.; investigation: I.C., V.E.-G. and J.A.A.-E.; resources: J.A.A.-E. and J.Y.; data curation: I.C.; V.E.-G. and J.A.A.-E.; writing—original draft preparation: I.C.; V.E.-G. and G.J.L.-M.; editing: I.C., J.Y. and G.J.L.-M.; visualization: G.J.L.-M. and V.E.-G.; supervision: J.Y. and G.J.L.-M.; project administration: J.Y. and J.A.A.-E.; funding acquisition for Peña Moñuz project: J.A.A.-E. All authors have read and agreed to the published version of the manuscript.

**Funding:** This research received no external funding. Only Idoia Claver has a Scholarship funded by the Community of Madrid by Order 3789/2022 with reference PIPF-2022/PH-HUM-25913 and by the European Union-Next Generation with reference CT19/23-INVM-130.

**Data Availability Statement:** Data are available in the bones and remains with tooth marks deposited in the Department of Prehistory, Ancient History, and Archaeology of the University Complutense Madrid (Spain).

**Acknowledgments:** This study has been carried out under the auspices of the project "Etnoarqueología en la Dehesa de Olmeda de Cobeta. Parque Natural del Alto Tajo (Guadalajara)", which is funded by the Junta de Castilla-La Mancha. Idoia Claver acknowledges her scholarship which is funded by the Community of Madrid by Order 3789/2022 with reference PIPF-2022/PH-HUM-25913 and by the European Union-Next Generation with reference CT19/23-INVM-130.

**Conflicts of Interest:** The authors declare no conflicts of interest.

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
