# Peer review of "Taphonomy as a Methodological Approach for the Study of Dog Domestication: Application to the Prehistoric Site of Peña Moñuz (Guadalajara, Spain)"

_heritage, doi:10.3390/heritage8010034_

Round 1
Reviewer 1 Report
Comments and Suggestions for Authors
What is not adequately explained in this or previous articles by these authors is details describing the database that forms the underpinning of this work. Were the animals used in these experiments hungry or expecting a meal when they were given the bovid bones? How long did they have access to the bones before they were collected? Did other carnivores potentially have access to the bones, and did extent of competition for the bones affect pit size? ? Does species (or individual) bodyweight or jaw size or tooth size (which tooth?) predict tooth pit size, thus suggesting that extinct species or those differing in size from the current species would leave tooth marks that could be identified by this methodology? How many different individual animals were used to produce the pits? I have little doubt that AI (or human) learning can adequately predict characteristics of the species that produced the pits, but I also suspect that a number of obvious features about a chewed bone might do the same and make the applicability of the method more broadly useful. As a minor note, if the authors think that bones heavily chewed by canids might indicate the domestication of dogs, they ought to say so directly and defend that assertion.
Author Response
Ref. 1. What is not adequately explained in this or previous articles by these authors is details describing the database that forms the underpinning of this work.
Answer: We thank and appreciate the reviewer’s comments regarding this article and for following our previous work. In our research, we are advancing gradually, aiming to propose new methodological developments while applying practical applications to archaeological sites. We will address the reviewer’s questions and incorporate some of these responses into the article to make it more comprehensive.
In line with the suggestions from the first and second reviewers, we have expanded the "Methods and Materials" section to enhance the information about the frameworks we are using.
Ref. 1: Were the animals used in these experiments hungry or expecting a meal when they were given the bovid bones? How long did they have access to the bones before they were collected?
Answer: In our experiments, we adhered to the feeding protocols of the parks, ensuring compliance with site-specific requirements. The animals were generally given bones after their meals. However, the bones provided had little meat to ensure that the animals left tooth marks on the bones.
The bones were typically left with each carnivore for one week or several days. In the case of hyenas, the protocol was adjusted because they consumed the bones within hours. Therefore, bones were removed from the hyenas three hours after they were given.
Did other carnivores potentially have access to the bones, and did extent of competition for the bones affect pit size? ?
Answer: No. Each carnivore was isolated in a specific enclosure, and each bone was consumed by a single individual. Thus, we can be certain that the same type of carnivore consistently leaves similar tooth marks (pits). This has also been verified by comparing marks made by different wolf packs from various locations. This consistency only applies to pits. In contrast, score marks may vary slightly between different populations of the same carnivore (e.g., wolves). For this reason, we focused exclusively on pits in this study.
Does species (or individual) bodyweight or jaw size or tooth size (which tooth?) predict tooth pit size, thus suggesting that extinct species or those differing in size from the current species would leave tooth marks that could be identified by this methodology?
Answer: Differences in size do not interfere with morphology. We conducted experiments with tigers and leopards and observed that sexual dimorphism did not influence the morphology of the marks. Pits produced by males and females were similar. These observations have been included in the manuscript. The size of the marks (i.e., their dimensions) only allows us to predict whether a carnivore produces large tooth marks (wolves, hyenas, bears, canids, jaguars, crocodiles) or small tooth marks (leopards, foxes, wildcats, cheetahs, pumas). However, it does not allow precise identification of the species. Combining this analysis with morphometric analysis increases precision and improves the ability to associate a pit with a specific carnivore.
Identifying which tooth makes the mark is far more complex because it requires isolating each movement involved in chewing and gripping the bones, which is difficult to analyze. There is always the possibility of multiple teeth being involved. Additionally, identifying the specific tooth is outside the scope of our research, as it is a question more relevant to veterinary studies. Our interest lies in identifying the carnivore species that left the marks, not the specific tooth responsible. To answer our research question, recognizing patterns that allow us to attribute the marks to a particular carnivore is sufficient.
How many different individual animals were used to produce the pits?
Answer: The answer to this depends on the type of carnivore. For example, the tooth marks produced by wolves were obtained from experiments with wolves in Cantabria, Cuenca, Asturias, and Zamora. Using a sample of over 500 wolf pits, we applied an algorithm to randomly select 50 marks, which were then compared with an equal number of marks produced by other carnivores involved in the study.
I have little doubt that AI (or human) learning can adequately predict characteristics of the species that produced the pits, but I also suspect that a number of obvious features about a chewed bone might do the same and make the applicability of the method more broadly useful.
Answer: In reality, most carnivores exhibit significant similarities in the way they modify bones. While hyenas chew bones more intensively than bears, lions, leopards, or canids, all these species generate marks through similar processes. For example, all of them occasionally produce cylindrical fragments (work in preparation) and may collapse a single epiphysis. They also all produce small tooth marks (less than 2 mm).
Consequently, we need tools to empirically differentiate, with minimal subjectivity, which carnivore is responsible for a given mark. Our methodology provides an effective tool for this purpose.
As a minor note, if the authors think that bones heavily chewed by canids might indicate the domestication of dogs, they ought to say so directly and defend that assertion.
Answer: The presence of heavily chewed bones at a site does not necessarily indicate the activity of domestic dogs. Other carnivores, such as foxes or wolves, can produce patterns similar to those of dogs. This underscores the importance of accurately distinguishing between them, particularly since both are canids. At the same time, dog domestication in late European prehistory is not really a controversial inference.
Reviewer 2 Report
Comments and Suggestions for Authors
This manuscript is highly innovative in that it attempts to provide insight into the domestication process by using indirect measures of taphonomic alterations to bone surfaces by toothmarks. The premise is that if one can identifying the specific carnivore gnawing on zooarchaeological faunal remains than it might be possible to identify the presence of early dog. If this approach is viable it would potentially be highly valuable to the discipline as a means for identifying the agent of bites marks commonly found at archaeological site and it would complement other methodologies used to explore the origins of dogs. As a result, this manuscript would likely be of great interest to the readers of Heritage. As discussed below, I believe there are a new presentation and methodological issues that needed to be addressed before this is ready for
Overall, this is a strong analysis built on several years of rigorous previous research by the authors and their colleagues (citations 43-47 & 67-70). This history of strong publication by the authors provide confidence in the results of this work; however, it also creates some of the most problematic aspects of this manuscript. First, there is limited discussion of the comparative database used as comparison, so the reader is forced to look at prior articles for background and understanding of the critical data being used in this study. More description of that dataset is need for the reader of Heritage to understand the results. Obviously, there is no need to thoroughly summarize each publication, but a concise discussion of key finding and results will allow the current reader to understand the key differences and expectations for the bite marks of these different comparative species would be useful to the reader.
My biggest concern of this manuscript arises from a related issues. The prior work of the authors resulted in a wonderful database of tooth marks from a diversity of species including hyena, jaguar, leopard, Lion and wild dog. As the authors had these data it was easy to makes comparisons with the Peña Moñuz species. However, the authors never seem to consider justifying why these species are included. Had this been a late Pleistocene site, I would not have any issue with their inclusion. Given we are talking about an Iron Age site in Spain, I am not certain these species were likely on the landscape in Iberia at that time. Excluding species that were either extinct or extirpated from Iberia 2000 years ago seems critical to the reliability of this study. Moreover, the results from figure 4 are obscured due to the inclusion of these irrelevant species. Reducing the comparison would likely make the results more relevant and obvious.
Just as importantly, one interesting comparison that is underdeveloped in this manuscript is the comparison of Peña Moñuz specimens to the know dogs breeds. Little discussion of the breed specific data is included here except for a throwaway line associated with Figure 5 that suggest these marks are not from Mastiffs. More thorough discussion and analysis of the breed data would be more informative. It would also potentially help make sense of the highly variable distribution of Canis familiaris. Do the many outlier data points make sense when you consider breed/body size differences.
This manuscript highlights the value of this methodology for exploring the origins of dog domestication. While I am not opposed to this suggestion, the current study actually contribute little to our understanding of dog domestication. As stated in the conclusion it is already well established that Peña Moñuz had dogs, and these dogs had been domesticated for more than 14000 year. Trying to explore this issue using an Iron Age site does not make sense. However, as discussed in the conclusion this methods is of great use in identifying which species might be within the Iron Age landscape and interacting with human. That is an important question—maybe not as big as dog domestication—and one that can be answered with the available data. This is a very strong analysis, but its scope is limited and there is no reason it cannot be discussed and appreciated for what it actually offers.
Finally, I feel it relevant to point out that the zooarchaeological focus of this appears to be better suited to other journals, e.g., Journal of Archaeological Method and Theory, Journal of Archaeological Science, Journal of Taphonomy, Advances in Archaeological Practice, and International Journal of Osteoarchaeology. Obviously, this decision is up to journal editor and authors. I just felt obliged to point this out.
Author Response
Ref. 2. This manuscript is highly innovative in that it attempts to provide insight into the domestication process by using indirect measures of taphonomic alterations to bone surfaces by toothmarks. The premise is that if one can identifying the specific carnivore gnawing on zooarchaeological faunal remains than it might be possible to identify the presence of early dog. If this approach is viable it would potentially be highly valuable to the discipline as a means for identifying the agent of bites marks commonly found at archaeological site and it would complement other methodologies used to explore the origins of dogs. As a result, this manuscript would likely be of great interest to the readers of Heritage. As discussed below, I believe there are a new presentation and methodological issues that needed to be addressed before this is ready for
Answer: Thank you very much for your comments and for considering this article. We have aimed to address the issues raised and respond to the questions. Accordingly, we have expanded the Results and Discussion section and added new information about the carnivores used in the experiments to the Materials and Methods section
Ref. 2. Overall, this is a strong analysis built on several years of rigorous previous research by the authors and their colleagues (citations 43-47 & 67-70). This history of strong publication by the authors provide confidence in the results of this work; however, it also creates some of the most problematic aspects of this manuscript. First, there is limited discussion of the comparative database used as comparison, so the reader is forced to look at prior articles for background and understanding of the critical data being used in this study. More description of that dataset is need for the reader of Heritage to understand the results. Obviously, there is no need to thoroughly summarize each publication, but a concise discussion of key finding and results will allow the current reader to understand the key differences and expectations for the bite marks of these different comparative species would be useful to the reader.
Answer: Thank you for the comment. Indeed, we can add a section providing a more detailed description of the comparative samples. Another reviewer also suggested this, so we have proceeded to include a section addressing this point.
Ref. 2. My biggest concern of this manuscript arises from a related issues. The prior work of the authors resulted in a wonderful database of tooth marks from a diversity of species including hyena, jaguar, leopard, Lion and wild dog. As the authors had these data it was easy to makes comparisons with the Peña Moñuz species. However, the authors never seem to consider justifying why these species are included. Had this been a late Pleistocene site, I would not have any issue with their inclusion. Given we are talking about an Iron Age site in Spain, I am not certain these species were likely on the landscape in Iberia at that time. Excluding species that were either extinct or extirpated from Iberia 2000 years ago seems critical to the reliability of this study. Moreover, the results from figure 4 are obscured due to the inclusion of these irrelevant species. Reducing the comparison would likely make the results more relevant and obvious.
Answer: We fully understand the reviewer’s concern. Including the different carnivores served as a control test or calibration for the study. If the Peña Moñuz marks had been associated with an extinct carnivore, it would have presented a significant interpretive challenge.
For this reason, it was important for us to compare the tooth marks of various carnivores with those from Peña Moñuz. However, it is true that it could be useful to add a new figure comparing Peña Moñuz with wolves and foxes to demonstrate that the marks do not correspond to these carnivores. We have implemented this suggestion as a new figure (Figure 5).
Ref. 2. Just as importantly, one interesting comparison that is underdeveloped in this manuscript is the comparison of Peña Moñuz specimens to the know dogs breeds. Little discussion of the breed specific data is included here except for a throwaway line associated with Figure 5 that suggest these marks are not from Mastiffs. More thorough discussion and analysis of the breed data would be more informative. It would also potentially help make sense of the highly variable distribution of Canis familiaris. Do the many outlier data points make sense when you consider breed/body size differences.
Answer: Following the suggestions, we have expanded the results by including new images and tables. Within this section, we have also incorporated additional discussion, along with new information presented as tables, figures, and a supplementary file.
Ref. 2. This manuscript highlights the value of this methodology for exploring the origins of dog domestication. While I am not opposed to this suggestion, the current study actually contribute little to our understanding of dog domestication. As stated in the conclusion it is already well established that Peña Moñuz had dogs, and these dogs had been domesticated for more than 14000 year. Trying to explore this issue using an Iron Age site does not make sense. However, as discussed in the conclusion this methods is of great use in identifying which species might be within the Iron Age landscape and interacting with human. That is an important question—maybe not as big as dog domestication—and one that can be answered with the available data. This is a very strong analysis, but its scope is limited and there is no reason it cannot be discussed and appreciated for what it actually offers.
Answer: We understand the reviewer’s perspective. However, we are starting from the premise that this methodology has never been applied to studying marks produced by extinct dogs. We know that dogs existed during the Iron Age, that they were most likely domesticated, but we do not know what types they were or whether their tooth marks were similar to those of modern dogs.
Our study begins with a working hypothesis, but we do not know whether it will hold true. The project starts with a prehistoric site to test whether our methodology can accurately identify dog activity in such contexts. Subsequently, we aim to apply the technique to older contexts, such as the Bronze or Copper Age, followed by Neolithic sites, and, ultimately, a Palaeolithic site. In this way, we intend to evaluate how effectively our method can analyze samples and determine whether it provides reliable results.
Ref. 2 Finally, I feel it relevant to point out that the zooarchaeological focus of this appears to be better suited to other journals, e.g., Journal of Archaeological Method and Theory, Journal of Archaeological Science, Journal of Taphonomy, Advances in Archaeological Practice, and International Journal of Osteoarchaeology. Obviously, this decision is up to journal editor and authors. I just felt obliged to point this out.
Answer: We agree with the reviewer that there are other journals where this work might also be suitable for publication. However, we chose Heritage for a number of reasons: a) we have not previously published there; b) it is a journal with significant impact, which is a critical requirement for one of the authors (IC); c) lastly, Heritage offers a faster publication process compared to other, slower journals.
Under normal circumstances, this would not be a concern, but this article will be part of IC’s thesis, and we prefer to avoid risks related to timing. Nonetheless, we appreciate the reviewer’s suggestions and will take them into account for future contributions.
Reviewer 3 Report
Comments and Suggestions for Authors
The article describes an innovative and impressive taphonomic method for the indirect analysis of the makers of the tooth holes in animal bones discovered at the prehistoric site of Peña Moñuz. Although researchers used a variety of advanced methods and methodologies, they needed the physical remains of dogs or wolves that created the bone assemblies. This work focuses on the application of high-resolution taphonomic analysis on bones modified by predators. The findings demonstrate the effectiveness of the technique for identifying the bite holes on the bones. and assist researchers in spatial analysis of the location and timing of the dog's domestication.
However, this work begins with the promise of clarifying who are the predators that left marks on the bone assemblage in Pena Moñuz, but in practice refrains from presenting an illustration of all the canines in the study area (jackal, fox, wolf and dog) in order to convince us beyond any doubt that the creators of the assemblage are domestic dogs.
However, this work begins with the promise of clarifying who are the predators that left marks on the bone assemblage in Pena Moñuz, but in practice refrains from presenting an illustration of all the canines in the study area (jackal, fox, wolf and dog) in order to convince us beyond any doubt that the creators of the assemblage are domestic dogs.
However, this work begins with the promise of clarifying who are the predators that left marks on the bone assemblage in Pena Moñuz, but in practice refrains from presenting an illustration of all the canines in the study area (jackal, fox, wolf and dog) in order to convince us beyond any doubt that the creators of the assemblage are domestic dogs.
In summary, this is a good and publishable work, yet several elements of the work should be improved:
1) Introduction - clear and well written.
2) Methods – detailed and clear, But the method of bones collecting on Peña Moñuz must be added in order to complete the taphonomic picture.
3) Results - detailed and well presented in tables and graphs.
However, the researchers clarified that the results clearly indicate canine teeth holes, therefore it is mandatory to show all the canines in the habitat (jackal, fox, wolf and dog) in a graph to clarify the differences between them and the prominence of the dog.
4) Discussion - It is appropriate and exhaustive on the one hand, but on the other hand it requires reference to the decision that the teeth of domestic dogs were embedded in the bones and not the teeth of other canines.
Author Response
Reviewer 3:
Ref 3. The article describes an innovative and impressive taphonomic method for the indirect analysis of the makers of the tooth holes in animal bones discovered at the prehistoric site of Peña Moñuz. Although researchers used a variety of advanced methods and methodologies, they needed the physical remains of dogs or wolves that created the bone assemblies. This work focuses on the application of highresolution taphonomic analysis on bones modified by predators. The findings demonstrate the effectiveness of the technique for identifying the bite holes on the bones. and assist researchers in spatial analysis of the location and timing of the dog's domestication.
However, this work begins with the promise of clarifying who are the predators that left marks on the bone assemblage in Pena Moñuz, but in practice refrains from presenting an illustration of all the canines in the study area (jackal, fox, wolf and dog) in order to convince us beyond any doubt that the creators of the assemblage are domestic dogs.
In summary, this is a good and publishable work, yet several elements of the work should be improved:
Answer: Thank you very much for your ratings and appreciations, we have taken your comments into account and followed all your suggestions, thank you.
- Introduction - clear and well written.
Answer: It is made, we have modified the introduction, now it is well written
- Methods – detailed and clear, But the method of collecting bones on Peña Moñuz must be added in order to complete the taphonomic picture.
Answer: It is made. All the aditional information have been added.
- Results - detailed and well presented in tables and graphs.
However, the researchers clarified that the results clearly indicate canine teeth holes, therefore it is mandatory to show all the canines in the habitat (jackal, fox, wolf and dog) in a graph to clarify the differences between them and the prominence of the dog.
Answer: We have added new figures, tables and a 2 new supplementary file
- Discussion - It is appropriate and exhaustive on the one hand, but on the other hand it requires reference to the decision that the teeth of domestic dogs were embedded in the bones and not the teeth of other canines
Answer: We have added new figures, tables and a 2 new supplementary file
Reviewer 4 Report
Comments and Suggestions for Authors
This manuscript presents good results from a narrow area of taphonomic research. I have a few questions that I hope the authors can address in a revision.
Which dog breeds should we expect to be most similar to the type/size of dog in the region during the Iron Age?
Ln 188. Clarify “other carnivores”. Other than dogs?
Ln 197. Why an Iron Age dog and not just one of the dog breeds in the modern sample?
How many of the measured pits came from the same chewed archaeological specimen? Is there much variation in pit shape and size within groups from the same bone samples?
Were human tooth pits considered? If not, please add an explanation for why they were not included.
Author Response
Rew 4.: This manuscript presents good results from a narrow area of taphonomic research. I have a few questions that I hope the authors can address in a revision.
Answer: Before addressing the reviewer’s questions, we would like to express our gratitude for reading this manuscript and considering it for potential publication. We will now respond to the specific points raised.
Rew 4. Which dog breeds should we expect to be most similar to the type/size of dog in the region during the Iron Age?
Answer: We do not have a definitive answer to this question, as little is known about dogs in the Iron Age of Celtiberia on the Iberian Peninsula. For our comparison, we chose medium-sized dogs because their size is similar to that of wolves, the primary wild carnivore whose bite marks could be confused with those of dogs. Additionally, we found it interesting to explore the possibility of mastiffs, as some authors have suggested that the mastiff breed might have already existed in inland Iberia during this period. However, our study of bite marks at this site did not provide evidence to support this hypothesis. It is worth noting that any discussion of dog breeds in the Iron Age will require significant multidisciplinary research to establish more concrete conclusions.
Rew 4. Ln 188. Clarify “other carnivores”. Other than dogs?
Answer: foxes, wolves
Rew 4Ln 197. Why an Iron Age dog and not just one of the dog breeds in the modern sample?
Answer: We have clarified the wording: “there appears to be a relationship between the tooth pits from Peña Moñuz and the marks typically associated with dogs”.
Rew 4. Is there much variation in pit shape and size within groups from the same bone samples?
Answer: In total, there are 15 bones with 26 bite marks. Eight bones exhibit a single pit each, five bones have two pits, one bone shows five pits due to furrowing, and one piece presents three pits. We have added this information to the manuscript.
Rew 4 Were human tooth pits considered? If not, please add an explanation for why they were not included.
Answer: Initially, we did not consider this possibility, as the marks in our study appeared to be associated with canines. However, following the reviewer’s suggestion, we included several comparative analyses in Figure 5, incorporating human dentition marks. These analyses confirmed that the marks are not human but were made by dogs.
Round 2
Reviewer 1 Report
Comments and Suggestions for Authors
The authors'replies to questions in previous reviews suggest that the protocalls for obtaining bones marked by carnivores varied in ways that probably affected the animals' behavior. Thus the value of the comparative sample is questionable. Also, the paper relies on other questionable assumptions: that all dogs in a particular breed are behaviorally and geometrically similar; that there is little or no variation in bone-chewing habits within a species; and that species' bone-chewing behaviors are distinctive and do not overlap. They also categorize dog breeds into different types without specifying what criteria are being used to define those types.
It is apparent that they have done a great deal of statistical and other work, but their presentation of the methodology and conclusions are unconvincing.
Author Response
Reviewer 1: Comments and Suggestions for Authors
The authors'replies to questions in previous reviews suggest that the protocalls for obtaining bones marked by carnivores varied in ways that probably affected the animals' behavior. Thus the value of the comparative sample is questionable.
Answer: We sincerely thank the reviewer for their time and thoughtful re-evaluation of our manuscript. However, we respectfully disagree with their assertion that the value of our samples is questionable due to alleged differences in the behavior of carnivores during the experiments. Below, we clarify our methods and the robustness of our dataset.
The tooth marks analyzed in this study were obtained under standardized experimental conditions. Carnivores—including jaguars, lions, wolves, bears, wild dogs, hyenas, and leopards—bit the diaphyses of long horse bones in captivity, ensuring consistency across species. The only exception was the wolf sample, which included marks produced by both captive and free-ranging wolves. However, extensive analyses of over 500 pits (e.g., Courtenay et al., ref. 48; Yravedra et al., ref. 46) have demonstrated that the captivity or freedom status of wolves does not influence the morphological or metrical characteristics of tooth pits. Specifically, ref. 48 compared four sites—two involving captive wolves and two involving free-ranging wolves—and found minimal variability in tooth mark morphology across these contexts.
Similarly, research on foxes has shown that prey size does not affect the morphology of tooth marks. For example, Courtenay et al. (ref. 68) demonstrated consistent tooth mark characteristics across prey ranging from cows to deer and goats. Furthermore, Herranz et al. (ref. 49) examined carnivores with pronounced sexual dimorphism and found no significant differences in tooth mark morphology between males and females, even when their body sizes varied greatly (e.g., tigers and leopards).
These findings collectively support the conclusion that intraspecific variability, captivity status, and prey size do not significantly affect tooth mark morphology or metrics. While studies on the spatial distribution of tooth marks have reported differences between wild and captive carnivores, our research does not analyze spatial patterns. Similarly, while scores can sometimes exhibit small population-level differences, they were not the focus of this study (see Courtenay et al., ref. 48).
Given the extensive research conducted over the past eight years, we are confident in the validity of our comparative dataset. We remain uncertain as to why the reviewer now raises these doubts, particularly given their prior comments indicating familiarity with and recognition of our work. We hope this explanation adequately addresses their concerns and reinforces the reliability of our experimental protocols and conclusions.
Reviewer 1: Also, the paper relies on other questionable assumptions: that all dogs in a particular breed are behaviorally and geometrically similar; that there is little or no variation in bone-chewing habits within a species; and that species' bone-chewing behaviors are distinctive and do not overlap. They also categorize dog breeds into different types without specifying what criteria are being used to define those types.
It is apparent that they have done a great deal of statistical and other work, but their presentation of the methodology and conclusions are unconvincing.
Answer: We appreciate the opportunity to address the reviewer’s concerns. To clarify, our study does not examine variations in how carnivores bite bones or the specific behaviors exhibited while doing so. Instead, our focus is on the final result of their behavior: the tooth marks left on the bones. These marks—whether produced by canines dragging along the surface, molars or premolars holding the bone, or through bone breakage under strong pressure—are the primary data we analyze.
Our research concentrates exclusively on pits, as these marks have consistently been shown to remain morphologically and metrically stable across different populations within the same species. This stability supports their use for comparative purposes and interspecies differentiation.
We do not delve into the behavior or mechanics of how marks are created; rather, our objective is to determine whether the morphology of tooth marks can reliably differentiate between species. For instance, our results successfully classified the tooth marks at Peña Moñuz as having been made by a dog.
The inclusion of multiple dog breeds in our study was intended to expand the comparative framework for distinguishing dog marks from those of wolves. As demonstrated in this and prior research, such differentiation is achievable. However, this work represents an initial effort, and we acknowledge the importance of continuing to expand the sample to include more dog breeds. This will allow us to further evaluate how variability among dogs affects tooth marks and to determine whether any overlap with wolves exists.
The broader aim of this research is to identify the presence of dogs through their tooth marks, particularly in archaeological contexts. While additional analyses (e.g., studying the specific marks made by individual teeth) could provide valuable insights, our study focuses on the more generalized patterns visible in the fossil record. Typically, these consist of elongated marks, circular pits, collapsed axial bones or epiphyses, and bones with varying numbers of marks. For example, felines often leave fewer marks per bone, while canines and hyaenids tend to leave more. The number of marks, however, can vary if multiple carnivores worked on the same bone.
By focusing on individual tooth marks rather than broader behavioral patterns, our methodology provides a robust means of classifying marks with a high degree of reliability. Moving forward, we hope this technique will prove sensitive enough to identify dog activity in earlier contexts, such as the Neolithic or Late Paleolithic. However, further application in these contexts is necessary to fully validate its utility.
We trust that this explanation clarifies our approach and methodology and demonstrates the potential of our research while addressing the reviewer’s concerns.
Reviewer 2 Report
Comments and Suggestions for Authors
The current review is the second review of this manuscript and as a result the focus of this review will be on how well the revised manuscript dealt with the original concerns. The original review was very supportive of this manuscript and thought that this research was highly innovative.
The original review identified only a few minor presentation and methodological issues that needed to be addressed before this is ready for publication. The original review suggested, 1) a need for more discussion of the comparative database used as comparison, 2) an opinion that certain comparative species used in this study, particularly hyena, jaguar, leopard, Lion and wild dog, were not necessary given the age of this site and their inclusion makes the results more difficult to interpret because of the additional of excessive “background noise”, and 3) a need for more thorough discussion and analysis of the tooth marks from different breed. The original review also identified some minor concerns that the focus on dog domestication did not accurately reflect the focus and analytical value of this paper and that the nature of this paper would perhaps be better suited to another journal.
Overall, the revisions to this manuscript are substantial and positively improve this document. In terms of my prior concerns of an inadequate discussion of methods and the comparative database are eliminate by the detailed discussion concerning the source of comparative samples added to section 2. While the authors choose to keep all the comparative species in the study—again making figure 4 difficult to interpret-- the authors did address my concern through the addition of a separate discussion on page 9 and inclusion of a new figure 5. These changes largely eliminate my earlier concerns. However, it should be noted that the new paragraph includes a statement (lines 294-295) in which the p value for the comparison of the Peña Moñuz canids marks and the comparative Canis familiaris sample is reported as p=.021; however, table Table 2 still reports that p value as .017. I believe the table needs to be updated. Also, the new figure 5 includes a comparison to humans bite marks and it might be helpful to have table 2 updated these results. One of the biggest improvement to the revised manuscript in my opinion is the substantial expansion of the comparison of the Peña Moñuz canids bit marks to the known breed marks. There was substantial discussion of these comparisons and addition of two new tables and a figure. While the modern known breeds did not exist during the Iron age, this analysis does provide a much more accurate estimate of the likely body size of the ancient dogs making these marks. Finally, the authors’ response to my minor concerns about focus on dog domestication and journal were adequate and these are no longer concerns for me.
In summary, this is a much improved manuscript and I think that this manuscript would likely be of great interest to the readers of Heritage and I think this articles will be highly valuable to the discipline.
Author Response
Reviewer 2:
Comments and Suggestions for Authors
The current review is the second review of this manuscript and as a result the focus of this review will be on how well the revised manuscript dealt with the original concerns. The original review was very supportive of this manuscript and thought that this research was highly innovative.
Answer: Thank you very much for your comments in the first review and this second review.
Reviewer 2: The original review identified only a few minor presentation and methodological issues that needed to be addressed before this is ready for publication. The original review suggested, 1) a need for more discussion of the comparative database used as comparison, 2) an opinion that certain comparative species used in this study, particularly hyena, jaguar, leopard, Lion and wild dog, were not necessary given the age of this site and their inclusion makes the results more difficult to interpret because of the additional of excessive “background noise”, and 3) a need for more thorough discussion and analysis of the tooth marks from different breed. The original review also identified some minor concerns that the focus on dog domestication did not accurately reflect the focus and analytical value of this paper and that the nature of this paper would perhaps be better suited to another journal. Overall, the revisions to this manuscript are substantial and positively improve this document. In terms of my prior concerns of an inadequate discussion of methods and the comparative database are eliminate by the detailed discussion concerning the source of comparative samples added to section 2. While the authors choose to keep all the comparative species in the study—again making figure 4 difficult to interpret-- the authors did address my concern through the addition of a separate discussion on page 9 and inclusion of a new figure 5. These changes largely eliminate my earlier concerns. However, it should be noted that the new paragraph includes a statement (lines 294-295) in which the p value for the comparison of the Peña Moñuz canids marks and the comparative Canis familiaris sample is reported as p=.021; however, table Table 2 still reports that p value as .017. I believe the table needs to be updated.
Answer: Indeed we forgot to update the latest version of table 2.
We have now updated the data in relation to the new analyses we did after the first round of revisions. Now text and table have the same data. Additionally, and to clarify figure 4, we have changed figure 4, classifying carnivores by groups and not by species.
Reviewer 2: Also, the new figure 5 includes a comparison to humans bite marks and it might be helpful to have table 2 updated these results. One of the biggest improvement to the revised manuscript in my opinion is the substantial expansion of the comparison of the Peña Moñuz canids bit marks to the known breed marks.
Answer: Table 2 has been updated to include data from humans,
Reviewer 2: There was substantial discussion of these comparisons and addition of two new tables and a figure. While the modern known breeds did not exist during the Iron age, this analysis does provide a much more accurate estimate of the likely body size of the ancient dogs making these marks. Finally, the authors’ response to my minor concerns about focus on dog domestication and journal were adequate and these are no longer concerns for me.
In summary, this is a much improved manuscript and I think that this manuscript would likely be of great interest to the readers of Heritage and I think this articles will be highly valuable to the discipline.
Answer: Thank you very much for your comments, they have been very constructive and have added value to the work. The use of pedigree dogs for our experiments is simply related to the fact of using known breeds of dogs as a frame of reference. Among the dogs used we have chosen medium breed dogs, and a mixed breed dog. We have to continue increasing the sample of dogs, adding smaller dogs, but this is a slow and progressive path that we hope to continue expanding, but we believe that medium breed dogs are the ones that most closely resemble the characteristics of wolves.
Reviewer 3 Report
Comments and Suggestions for Authors
The article is innovative and interesting and its results are clear and fascinating. The article deserves to be published in this format.
Author Response
Reviewer 3:
The article is innovative and interesting and its results are clear and fascinating. The article deserves to be published in this format.
Answer: Thank you very much for your comments. We are glad to know that you liked our study. We hope that this is the first work in a new line of research.
Round 3
Reviewer 1 Report
Comments and Suggestions for Authors
In your opening paragraph, you suggest that your method can be used without access to fossil specimens. That is true ONLY IF you have access to your high quality photographs of all surfaces of the specimens, which in turn requires access to the specimens or a collaborator who does have access to the original specimens. My main problem is that your comparative sample was obtained under different conditions which might impact the target animals’ chewing behavior. Another issue is the presence of young carnivores whose mouths would be smaller and that might have a non-adult dental array. Another potential issue is whether or not competing species are present. Finally, you don’t deal with the issue that extinct species might have different behaviors or different dental arrays.
Your basic premise seems to be that you can identify the species responsible for the damage to bones by measuring the pits on the bones, but different faunas and different circumstances cast doubt on your conclusions. I can see you have done a great deal of work, but animal behavior is too variable to support your conclusions
Author Response
Reviewer 1 Comment: In your opening paragraph, you suggest that your method can be used without access to fossil specimens. That is true ONLY IF you have access to high-quality photographs of all specimen surfaces, which in turn requires access to the specimens or a collaborator who does have access to the original specimens.
Reply:
This assertion is incorrect. As we demonstrate in the article, our experimental samples were obtained using captive carnivores. These samples serve as our reference framework until we can obtain comparable data from wild carnivores. Studies comparing captive and wild carnivores (e.g., Sala et al., 2014; Gidna et al., 2013; Mora et al., 2022) reveal differences in the distribution, location, and frequency of tooth marks. However, these differences do not affect the morphometry of tooth marks.
In fact, Courtenay et al. (2023) provides comprehensive evidence showing no morphometric differences between tooth marks produced by captive and wild carnivores. Marks from both populations are classified in the same way. Therefore, the morphometric framework based on captive carnivores is fully applicable to marks left by the same carnivore species in the wild.
References:
Sala et al., 2011 Sala, N., Arxsuaga, J. L. Haynes, G. 2014. Taphonomic comparison of bone modifications caused by wild and captive wolves (Canis lupus). Quaternary International Volume 330, 30 April 2014, Pages 126-135
Gidna et al., 2014 Gidna, A., Yravedra, J., Domínguez Rodrigo, M. 2013. A cautionary note on the use of captive carnivores to model wild predator behavior: a comparison of bone modification patterns on long bones by captive and wild lions. Journal of Archaeological Science 40, 1903-1910
Mora et al., 2022 Mora, R.; Aramendi, J.; Courtenay, L.A.; González-Aguilera, D.; Yravedra, J.; Maté-González, M.Á.; Prieto-Herráez, D.; Vázquez-Rodríguez, J.M.; Barja, I. Ikhnos: A Novel Software to Register and Analyze Bone Surface Modifications Based on Three-Dimensional Documentation. Animals 2022, 12, 2861. https://doi.org/10.3390/ani12202861
Courtenay et al., 2023 Courtenay, L. A.; Herranz-Rodrigo, D.; Yravedra, J.; Vázquez-Rodríguez, J. M.; Huguet, R.; Barja, I.; Maté González, M. A.; Fernández, M.; Muñoz-Nieto, A. L.; González-Aguilera, D. 3D Insights into the Effects of Captivity on Wolf Mastication and Their Tooth Marks; Implications in Ecological Studies of Both the Past and Present. Animals, 2021, 11(8), 2323
Reviewer 1 Comment: Another issue is the presence of young carnivores whose mouths would be smaller and that might have a non-adult dental array.
Reply:
This argument is speculative. The key issue is not the size of the carnivore's mouth but whether juveniles have deciduous dentition compared to the permanent dentition of adults. Regardless, this point is irrelevant.
When we compared tooth marks produced by captive carnivores (all adults) with those from wild carnivores (including both adults and juveniles), we found no morphometric differences between marks produced by adults and juveniles.
Additionally, when examining size-related issues in dimorphic carnivores, where males are significantly larger than females (e.g., leopards or tigers), our data indicate no morphometric differences in tooth marks between males and females. Therefore, as evidenced by Herranz et al. (2021), size is not a variable that affects the morphometry of tooth marks, contrary to the reviewer’s suggestion.
Reference:
Herranz et al., 2021 Herranz-Rodrigo, D.; Tardáguila-Giacomozzi, S.J.; Courtenay, L.A.; Rodríguez-Alba, J.-J.; Garrucho, A.; Recuero, J.; Yravedra, J. New Geometric Morphometric Insights in Digital Taphonomy: Analyses into the Sexual Dimorphism of Felids through Their Tooth Pits. Appl. Sci. 2021, 11, 7848. https:// doi.org/10.3390/app11177848
Reviewer 1 Comment: Another potential issue is whether or not competing species are present.
Reply:
This statement is also speculative. As the reviewer is aware, all social carnivores live in competitive environments. When wolves or hyenas access a carcass, they compete with their packmates for access. However, this does not influence the morphometry of pits.
As shown in Courtenay et al. (2023), pits are the only type of tooth mark that remain consistent across individuals of the same species. Conversely, other types of marks (e.g., scores) can vary morphologically depending on stress levels or other factors.
In Courtenay et al. (2023), we demonstrated that pits from four different wolf populations were morphometrically consistent under all conditions, whereas scores exhibited variability. For this reason, we focus exclusively on pits and do not use scores in our analysis.
This explanation was provided in prior review rounds.
Reference:
Courtenay et al., 2023 Courtenay, L. A.; Herranz-Rodrigo, D.; Yravedra, J.; Vázquez-Rodríguez, J. M.; Huguet, R.; Barja, I.; Maté González, M. A.; Fernández, M.; Muñoz-Nieto, A. L.; González-Aguilera, D. 3D Insights into the Effects of Captivity on Wolf Mastication and Their Tooth Marks; Implications in Ecological Studies of Both the Past and Present. Animals, 2021, 11(8), 2323
Reviewer 1 Comment: Finally, you don’t deal with the issue that extinct species might have different behaviors or different dental arrays.
Reply:
This comment is speculative. In fact, this issue has been addressed in previous studies, including Yravedra et al. (2022) and Courtenay et al. (2022b, 2023).
Regarding the current article, this is not a concern because the carnivores from the Iron Age (e.g., dogs, bears, foxes, wolves) are the same species that exist today. If the reviewer has doubts about the principles of actualism and uniformitarianism, that would be a separate discussion beyond the scope of this article.
References:
Yravedra et al., 2022 Taphonomic characterisation of tooth marks of extinct Eurasian carnivores through GeometricMorphometric. Science Bulletin, DOI: 10.1016/j.scib.2022.07.017
Courtenay et al., 2022 Courtenay, L.; Yravedra, J.; Herranz-Rodrigo, D.; Rodríguez, J.J.; Serrano-Ramos, A.; Estaca-Gómez, V.; Gonzálo-Aguilera, D.; Solano, J.A.; Jiménez-Arenas, J.M. Deciphering carnivoran competition for animal resources at the 1.46 Ma early Pleistocene site of Barranco León (Orce, Granada, Spain), 2022, 300, 107912
Courtenay et al., 2023 Courtenay, L. A.; Herranz-Rodrigo, D.; Yravedra, J.; Vázquez-Rodríguez, J. M.; Huguet, R.; Barja, I.; Maté González, M. A.; Fernández, M.; Muñoz-Nieto, A. L.; González-Aguilera, D. 3D Insights into the Effects of Captivity on Wolf Mastication and Their Tooth Marks; Implications in Ecological Studies of Both the Past and Present. Animals, 2021, 11(8), 2323
Reviewer 1 Comment:
Your basic premise seems to be that you can identify the species responsible for the damage to bones by measuring the pits on the bones, but different faunas and different circumstances cast doubt on your conclusions. I can see you have done a great deal of work, but animal behavior is too variable to support your conclusions.
Reply:
This is the reviewer’s opinion. While it is true that carnivore behavior can be variable, as we have demonstrated in more than a dozen studies (and as corroborated by other researchers), the morphometry of pits does not reflect that behavioral variability. Statistical analyses consistently show high resolution in distinguishing pits produced by different species, including dogs, wolves, African wild dogs (Lycaon), foxes, leopards, tigers, jaguars, lions, pumas, crocodiles, bears, and hyenas. These pits can be classified with over 90% accuracy.
Additionally, in studies applied to fossil records, it has been possible to distinguish tooth pits of Crocuta from Pachycrocuta, as well as lion pits from Homotherium pits (references available).
If the reviewer has evidence demonstrating that this classification system lacks resolution, we encourage them to present it in a publication.
Furthermore, these classification systems can be approached using various methodologies, including geometric morphometry combined with robust statistics (as in this paper), machine learning, deep learning, and even Fourier theorem applications